# Pathways, Processes, and Candidate Drugs Associated with a *Hoxa* Cluster-Dependency Model of Leukemia

**DOI:** 10.3390/cancers11122036

**Published:** 2019-12-17

**Authors:** Laura M. Kettyle, Charles-Étienne Lebert-Ghali, Ivan V. Grishagin, Glenda J. Dickson, Paul G. O’Reilly, David A. Simpson, Janet J. Bijl, Ken I. Mills, Guy Sauvageau, Alexander Thompson

**Affiliations:** 1Centre for Cancer Research and Cell Biology, Queen’s University Belfast, Belfast BT9 7AE, UK; lkettyle01@qub.ac.uk (L.M.K.); grishagin@gmail.com (I.V.G.); paul.oreilly@philips.com (P.G.O.); k.mills@qub.ac.uk (K.I.M.); 2Maisonneuve-Rosemont Hospital Research Center, Montréal, QC H1T 2M4, Canada; charles-etienne.lebert-ghali@umontreal.ca (C.-É.L.-G.); janettabijl@gmail.com (J.J.B.); 3Department of Microbiology, Infectious Disease and Immunology, Université de Montréal, Montréal, QC H3A 2B4, Canada; 4King’s College London, Division of Cancer Studies, London WC2R 2LS, UK; glenda.dickson@kcl.ac.uk; 5Centre for Experimental Medicine, Queen’s University Belfast, Belfast BT9 7BL, UK; David.Simpson@qub.ac.uk; 6Department of Medicine, Université de Montréal, Montréal, QC H3G 2M1, Canada; guy.sauvageau@umontreal.ca; 7Division of Hematology, Maisonneuve-Rosemont Hospital, Montreal, QC H1T 2M4, Canada; 8Institute for Research in Immunology and Cancer, Université de Montréal, Montréal, QC H3T 1J4, Canada; 9University of Nottingham Biodiscovery Institute, Division of Cancer and Stem Cells, University of Nottingham, Nottingham NG7 2UH, UK

**Keywords:** *HOXA* cluster, *MLL*-rearrangement, conditional deletion, leukemia maintenance, gene signature, pathways, candidate drugs

## Abstract

High expression of the *HOXA* cluster correlates with poor clinical outcome in acute myeloid leukemias, particularly those harboring rearrangements of the mixed-lineage-leukemia gene (*MLLr*). Whilst decreased *HOXA* expression acts as a readout for candidate experimental therapies, the necessity of the *HOXA* cluster for leukemia maintenance has not been fully explored. Primary leukemias were generated in hematopoietic stem/progenitor cells from *Cre* responsive transgenic mice for conditional deletion of the *Hoxa* locus. *Hoxa* deletion resulted in reduced proliferation and colony formation in which surviving leukemic cells retained at least one copy of the *Hoxa* cluster, indicating dependency. Comparative transcriptome analysis of *Hoxa* wild type and deleted leukemic cells identified a unique gene signature associated with key pathways including transcriptional mis-regulation in cancer, the Fanconi anemia pathway and cell cycle progression. Further bioinformatics analysis of the gene signature identified a number of candidate FDA-approved drugs for potential repurposing in high *HOXA* expressing cancers including MLLr leukemias. Together these findings support dependency for an *MLLr* leukemia on *Hoxa* expression and identified candidate drugs for further therapeutic evaluation.

## 1. Introduction

The mixed lineage leukemia gene (*MLL*) encodes a 431 kD multifunctional protein recently re-designated as lysine-specific methyltransferase 2A (KMT2A). Gene rearrangements involving *MLL*, termed *MLLr*, are among the most potent oncogenic drivers of leukemia detected in over 70% of acute lymphoblastic (ALL), up to 50% of acute myeloid leukemia (AML) cases in infants [1,2,3,4] and approximately 10% of adult leukemia including refractory therapy-related cases [5,6,7,8]. Currently, there are no specific therapies approved for *MLLr* and identifying target and pathway dependencies is critical to improving clinical outcome for these patients.

The *Class I homeobox* (*Hox*) genes are well established as downstream targets of MLL, conserved from the ancestral *Drosophila melanogaster* orthologs *HOM-C* complex and *Trithorax* respectively [9,10,11,12,13]. In conjunction with polycomb repressor complex (PRC) proteins, MLL has an essential role in embryogenesis and definitive hematopoiesis through maintenance of *Hox* gene expression and instigation of progenitor cell proliferation and differentiation [14,15,16]. Complete knockout of *Mll* is embryonic lethal in mice due in part to loss of *Hoxa7*, *Hoxa9,* and *Hoxc9*, and heterozygous *Mll*^wt/-^ exhibit hematopoietic abnormalities, retarded growth and deformities of the axial skeleton due to dysregulated *Hox* expression [13,17]. Mll was also shown to be required for normal hematopoietic stem and progenitor cell (HSPC) activity in conditional knock-out models [18,19]. In human, high expression of *HOXA* cluster genes, is a hallmark of high-risk, refractory AML, particularly *MLLr* leukemias [16,20,21,22,23,24] and direct binding of MLL-fusion proteins to *HoxA* promoter regions results in increased expression of these genes in leukemic models [25].

Whilst several studies indicate a need for *Hoxa* expression, primarily *Hoxa9*, in the establishment of *MLLr* leukemia their absolute requirement for disease progression and maintenance is less clear [26,27,28]. To address this, a *MLL-AF9* AML transplantation model (MA9) was generated in a previously reported conditional *Hoxa* cluster (*Hoxa*^flox/flox^) background [29]. Homozygous deletion of the *Hoxa* cluster was not tolerated by MLL-AF9. However, significant reduction in expression of *Hoxa7*, *Hoxa9*, *Hoxa10,* and *Hoxa11* was demonstrated and a *Hoxa*^del^ gene signature including upregulation of *Mpo*, *Clk-1*, *Clk-4*, and *Ccl6* and downregulation of *Angpt1*, *Emerin*, *Ddx6*, and *IL31ra* was obtained. Gene set enrichment and associated bioinformatics analysis of the *Hoxa*^del^ signature identified enriched processes including transcriptional repressor activity, myeloid, monocytic and leukemia differentiation, and canonical pathways including transcriptional mis-regulation in cancer and the Fanconi anemia pathway. Further bioinformatics analysis using connectivity mapping identified candidate drugs for potential repurposing in HOXA expressing cancer models including *MLLr* and associated leukemias.

## 2. Results

### 2.1. Establishment and Validation of Conditional Hoxa Leukemia Models

Fresh HSPCs enriched from bone marrow of *Hoxa*^flox/flox^ (AFF), MxCre^+^/*Hoxa*^flox/flox^ (MAFF), or control CD45.1 mice were transduced with *MA9* to generate leukemias. Extended serial re-plating in methylcellulose selected for transformed HSPCs with high proliferation potential. Condensed granulocyte-macrophage colonies (CFU-GM) were produced from *MA9* transduction and serial re-plating resulted in increased condensation of the colony (Figure 1A). Single colonies obtained from P3 cultures were used to generate cell lines. Transformed cells (P3–P5) were subsequently transplanted into sub-lethally irradiated recipient mice to generate leukemias. Recipient mice developed primary leukemias within reported time frames (Figure 1B) and immunophenotypes (Appendix A). Secondary leukemias, generated from direct transplantation of primary leukemias, were more aggressive with all mice succumbing to death within 50 days accompanied by tissue infiltration and splenomegaly (Figure 1C,D). Comparative gene expression analysis demonstrated increased *Hoxa* expression in all MA9 leukemias generated, compared to normal bone marrow (NBM), independent of the genetic background (Figure 2).

### 2.2. Reduced Leukemia Colony Formation Following IFNα-Induced Hoxa Deletion

Direct incubation with interferon-alpha (IFNα) activated *Cre* in the *Mx-1* background only and resulted in visible reduction in MAFF-MA9 colony formation compared to control CD45.1-MA9 cells or MAFF-derived NBM (Figure 3A upper panel). Direct colony counts demonstrated significant reduction in the number of colonies observed in MAFF-MA9 cells compared to PBS control and no measurable IFNα toxicity at the concentrations used (1U and 2.5U) in MAFF-derived NBM cells (Figure 3A lower panel).

PCR analysis of gDNA obtained from MAFF-MA9 cells showed levels of the *Hoxa*^del^ amplicon equivalent with the *Hoxa*^wt^ amplicon, presumably due to background *Cre* activation that was not markedly altered by IFNα treatment in bulk cells (Figure 3B, left panel). To explore the possibility that *Hoxa*^del/del^ cells were out-competed by their *Hoxa*^del/wt^ counterparts in bulk cultures, 100 individual MAFF-MA9 colonies from each treatment were picked and assayed by PCR. All 100 colonies demonstrated retention of at least one copy of the *Hoxa* cluster that had been unaffected by IFNα-treatment (sample gel Figure 3B, right panel). Isolation of the *Hoxa*^del^ amplicon followed by Sanger sequencing confirmed deletion of the loci spanning ~100 kb (Figure 3C and Appendix A).

### 2.3. Extension in Survival Following In Vivo Deletion of Hoxa Cluster

To examine whether propagation of the leukemia was dependent on maintained *Hoxa* expression engrafted mice that received IFNα- or PBS-treated MAFF-MA9 leukemic cells were further treated in vivo with Poly I:C (Figure 4A). A significant increased survival following Poly I:C treatment was observed compared to PBS controls (Figure 4B). All control mice succumbed to disease by 21 days, whereas Poly I:C treated mice survived up to day 26, indicative of reduction in number or function of leukemia repopulating cells (LRCs) due to in vivo *Hoxa* deletion. A significant increase in survival was also observed in cells that had been pre-treated, *ex vivo*, with IFNα compared to PBS (Figure 4B). Additional treatment of these cells with Poly I:C did not further increase survival suggesting that LRCs that escape *Hoxa* deletion are refractory to further Cre-treatment. In addition, reduced leukemia burden was observed in an independent (MAFF-MA9-Luc+) cohort of transplanted NOD-scid IL2rγnull (NSG) mice using bioimaging (Figure 4C). Recipient mice that succumbed to leukemia from either treatment arm retained the *Hoxa*^wt^ allele (Appendix A).

### 2.4. Hoxa^del^ Signature and Associated Pathways

Due to the potential loss of *Hoxa*^del^ cells over time, Illumina BeadArray-based gene expression profiling was performed on freshly sorted Cre-GFP and GFP-control treated AFF-MA9 cells. Methylcellulose-based colony formation and PCR demonstrated efficient deletion of the *Hoxa* locus (Appendix A). Overall output was characterized by plotting significance in the form of negative log10 (Adjusted *p*-value) versus log2 expression (fold change) for each gene (Figure 5). Genes with -log10 (Adjusted *p*-value) > 2 and log2 (fold change) > 0.5 were considered differentially expressed. Transcripts induced or repressed at log2 fold change ≥ 0.5; *p* ≤ 0.05 were sorted in ascending order by adjusted P-value. One hundred and fifty three probes, representing 135 differentially expressed genes (Appendix A) were identified for further bioinformatics evaluation.

### 2.5. Gene Set Enrichment Analysis

To identify candidate pathways, ontologies and drug interactions, differentially expressed genes were further analyzed by Enrichr [30,31] against curated data sets. In total, 151 probes representing 135 transcripts (123 increased expression, 12 decreased expression) met the criteria for analysis (log2 fold change ≥ 0.5; *p* ≤ 0.05). A summary of the analysis indicates a role for the *Hoxa* cluster in key biological processes and pathways (Figure 6). Cross reference of the *Hoxa*^del^ signature to the NCBI drug signatures database for gene set analysis (DSigDB) identified several candidate drugs, previously shown to have reported effects in other cancers, for repurposing in HOXA-associated leukemia.

## 3. Discussion

The clinical impact of targeted therapy against oncogenic dependency is exemplified by successful treatment of chronic myeloid leukemia (CML) with tyrosine kinase inhibitors (TKIs). In a large cohort study (n = 2662) life expectancy of CML patients that respond to TKIs was reported to approach the life expectancy of the general population [32]. Similar approaches to develop effective targeted therapies for other cancers are ongoing for which identification of oncogenic dependency pathways and processes is a prerequisite.

The association of deregulated *HOX* gene expression with cancer is well established (reviewed by Grier et al. [33]) particularly in aggressive forms such as acute leukemias (reviewed by Alharbi et al. [34]) in which the *Hoxa* cluster is predominant. Loss- and gain-of-function studies have identified roles for individual *Hoxa* genes with normal hematopoiesis including regulation of lineage commitment [35,36], development [37,38], self-renewal, and stem cell expansion [39,40,41]. However, the need for continuous *HOXA* expression in leukemia initiation and progression remains unclear [26,28,42,43]. This is due in part to functional redundancy within the *HOX* network and the potential oncogenic roles of non-coding elements within the *HOXA* locus including *HOTTIP* [44] and *HOXA-AS2* [45].

Herein, we demonstrate the criticality of the *Hoxa* cluster in the maintenance of established and aggressive MA9 leukemia in a murine model. Conditional *Cre*-recombinase based deletion of the *Hoxa* locus resulted in loss of leukemic phenotype in vitro and extension of survival in vivo. This phenotype was associated with differential expression of the *Hoxa* cluster, primarily *Hoxa7*-*Hoxa11*. Resistant leukemias and colonies retained one allele of the *Hoxa* locus, instead of upregulating paralogs, reflective of an ‘escapee’ rather than a ‘functional redundancy’ phenotype. Transplantation of *Hoxa*^del^ cells resulted in extension in time to leukemia. In contrast to the normal hematopoiesis setting, which tolerates biallelic deletion of the *Hoxa* locus, albeit with reduced HSPC proliferation and self-renewal [46] biallelic deletion was not tolerated in MA9 cells and all subsequent single-cell generated colonies retained at least one allele of the *Hoxa* locus. Together, this demonstrates unique dependency on a subset of the *Hoxa* cluster (*Hoxa7*, *Hoxa9*, *Hoxa10,* and *Hoxa11*) genes for maintenance of MA9 leukemia.

In association with the well-defined role for *HOX* in transcription regulation, hematopoietic stem cell self-renewal and maturation, several zinc finger, histone, cell cycle (e.g., *Clk-1*, *Clk-4*) and blood cell marker (e.g., *Mpo*) genes were over expressed following *Hoxa* cluster deletion. In combination with *Hoxa7*, *a9*, and *a10*, reduced expression of endothelial/adhesion molecules (*Heg1* and *Angptl1*) may be associated with altered *Hoxa*^del^ cell-niche interactions.

Bioinformatics analysis of the *Hoxa*^del^ gene signature identified significant enrichment of biological processes associated with blood cell differentiation, nucleosome association and chromatin assembly along with molecular functions including transcription repressor activity. The key pathway associated with the *Hoxa*^del^ signature was transcriptional misregulation in cancer. Several drugs with anti-cancer properties reported for solid tumors including, Mefloquine in glioblastoma [47], Digitoxigenin in renal carcinoma [48] and Emetine in bladder cancer [49] were associated with the *Hoxa*^del^ signature. Furthermore, some of these drugs e.g., Emetine and Cephaline were recently shown to be highly active against primary chronic lymphocytic leukemia cells by repressing HIF-1α and disturbing intracellular redox homeostasis [50].

The most represented (*n* = 3) compound in the top 10 DSigDB drugs associated with the *Hoxa*^del^ signature, Anisomycin, was recently identified as an expression mimic of mutant *RUNX-1* (*mt-RUNX-1*) knockdown with specific potency against cells from AML patients with germline or somatic *mt-RUNX-1* [51]. This is of particular interest as wild type RUNX-1 is essential for the maintenance of MLL-AF9 leukemia [52]. It is intriguing to postulate that upregulation of the *HOXA* cluster, central to progression in the most refractory AML subtypes and aggressive forms of cancer, is potentially druggable by such loss of expression mimics. Further analysis of the efficacy of such drugs across cell lines and patient samples in conjunction with toxicity profiles against normal cells is warranted to identify and de-risk candidates for potential clinical application.

## 4. Materials and Methods

### 4.1. Mouse Strains

*Hoxa*^flox/flox^ [46], *MxCre*, C57Bl/6-Ly5.1/CD45.1, C57Bl/6J/CD45.2 (Charles River, Harlow, UK) and generated compound *MxCre*^+^/*Hoxa*^flox/flox^ (MAFF) mice were maintained in SPF facilities under guidelines of The UK Animals (Scientific Procedures) Act 1986. Experimental procedures under PPL2760 were approved by the QUB Ethical Review Board and age-matched randomized animals were used throughout. Genotyping of litters was achieved by PCR (Appendix A).

### 4.2. Generation of Primary Leukemias and Cell Lines

Mouse hematopoietic stem and progenitor cells (HSPCs) isolated from bone marrow (Cat# 19856, EasySep™, STEMCELL Technologies, Cambridge, UK) were pre-stimulated overnight in RPMI1640/10% FCS/2 mM L-Glutamine (Life Technologies, Paisley, UK) supplemented with 10 ng/mL each of IL-3, IL-6, GM-CSF and 100 ng/mL SCF. Pre-stimulated HSPCs were transduced with MSCV-based retroviral (MLL-AF9, Cre-GFP and GFP control) or SFFV-based lentiviral supernatants (pSLIEW) as previously described [53,54]. Transduced HSPCs (1 × 10^4^ cells) seeded in Methocult M3434 (Stem Cell Technologies, Cambridge, UK) and re-plated every 6 days. Primary clonal cell lines were generated from third round re-plating and cultured in RPMI media containing 5 ng/mL each of IL-3, IL-6, GM-CSF, and 50 ng/mL SCF. Primary leukemias developed in sub-lethally (450–600 cGy) irradiated recipient mice were immunophenotyped by flow cytometry analysis (BD LSR II platform; BD Biosciences, Indianapolis, IN, USA) using established protocols and antibodies.

### 4.3. Deletion of the Hoxa Cluster

In vitro deletion of the *Hoxa* locus was by IFNα (R&D Systems, Abingdon, UK) treatment at the indicated dose and time. In vivo deletion in MAFF-MA9 leukemic mice was achieved by intraperitoneal (IP) injection of Polyinosinic:polycytidylic acid (Poly I:C; GE Healthcare Life Sciences, Buckinghamshire, UK) as previously described [46]. Briefly, mice were given 10 µg/g Poly I:C or vehicle (PBS) up to a maximum of 250 µg/mouse. Ex vivo deletion was by direct exposure to MSCV-Cre-GFP or MSCV-GFP control retroviral supernatants followed by cell sorting and gene expression profiling. Mice transplanted with traceable (pSLIEW) leukemias were injected IP with 150 mg D-luciferin/kg (Gold Biotechnology, St. Louis, MO, USA) and imaged using the Xenogen IVIS 200 (PerkinElmer, Buckinghamshire, UK). Nested PCR was used to identify individual colony bands.

### 4.4. Gene Expression Analysis and Bioinformatics

Total RNA was isolated using Trizol^®^ (Life Technologies), and its quality was assessed using Agilent RNA 6000 Nano chip and Agilent 2100 Bioanalyzer (Agilent Technologies, Santa Clara, CA, USA) according to the manufacturer’s instructions. RNA integrity (RIN) values of > 9 were obtained for all samples prior to application to the Mouse Ref-8 expression v.2 BeadArray (Illumina, San Diego, CA, USA), according to the manufacturer’s instructions and as previously reported [55]. Briefly, 550 ng of each RNA was reverse transcribed then in vitro transcribed to generate biotinylated cRNAs. Aliquots of labelled cRNAs (750 ng) were then hybridized to the BeadArray for 16–18 h at 58 °C. Signal detection was achieved using Amersham fluorolink streptavidin-Cy3 (GE Healthcare Bio-Sciences, Little Chalfont, UK) and scanned images obtained using the Illumina BeadArray confocal scanner. Bead level data of biological triplicates were transformed to log2 scale, normalized (quantile method) using beadarray R package [56] and annotated (Gene Symbols, Names, Entrez IDs, and Probe Quality Grades) using illuminaMousev2.db R package. Probes with poor quality grades (and Pgk-1 positive control for *Cre*-treatment) were removed. Differential expression was assessed by linear regression followed by parametric empirical Bayes analysis using limma R package [57] and output obtained by plotting significance versus log2 expression (fold change). Genes with expression ratios above log2 (0.5)-fold between treatments and control (*p* < 0.05 and *p* < 0.01) were identified. False discovery rates (FDR) were controlled using the Benjamini-Hochberg algorithm. Hierarchical clustering with complete linkage and Euclidean distance was performed. The Hoxa^del^ signature obtained was submitted as a fuzzy list to the Enrichr platform [30,31] to identify association with 156-curated libraries. Outputs were tabulated to show the Combined Score as a function of the log Fisher exact test *p*-value and *z*-score for deviation from expected rank plotted against the key pathway, process or drug interactions.

### 4.5. Statistical Analysis

Student’s t-tests were performed using GraphPad Prism software v7.0 (GraphPad Software, La Jolla, CA, USA). Means ± SEM were plotted and significance denoted by the value or as * *p* ≤ 0.05; ** *p* ≤ 0.01, *** *p* ≤ 0.001 throughout.

## 5. Conclusions

In conclusion, our findings directly demonstrate, for the first time, absolute dependency on the *Hoxa* locus for the maintenance of MA9 leukemia and identifies molecular processes and key pathways for further analysis and candidate drugs for pre-clinical validation and redeployment to this highly refractory disease. Several of the candidates identified here, by association with a *Hoxa*^del^ phenotype, have demonstrated efficacy in a variety of cancer subtypes. Extension of these findings to other *MLLr* leukemias or cancers associated with high *HOXA* expression warrants further investigation.

## Figures and Tables

**Figure 1 cancers-11-02036-f001:**
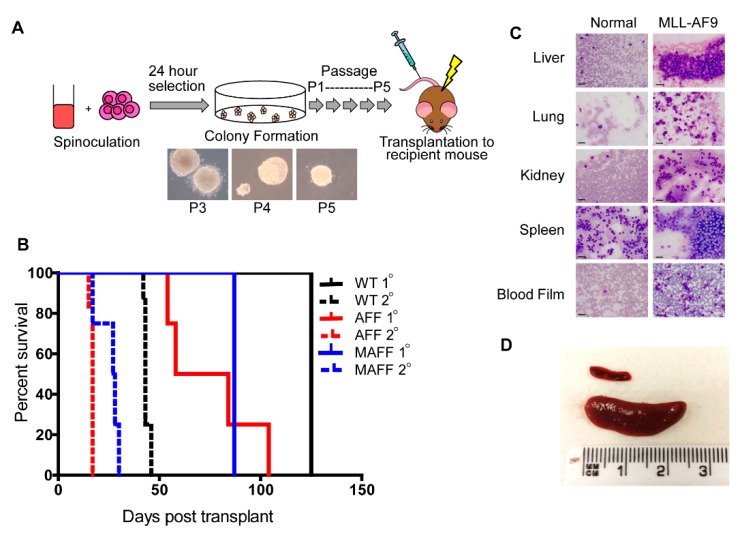
Development of MLL-AF9 (MA9) leukemias. (**A**) Donor hematopoietic stem/progenitor cells (HSPCs) were spinoculated with *MA9* retroviral particles and serially passaged in methylcellulose prior to transplantation into recipient mice. (**B**) Kaplan–Meier plot demonstrating survival of transplanted mice receiving MA9 leukemic cells derived from *Hoxa*^flox/flox^ (AFF), MxCre^+^/*Hoxa*^flox/flox^ (MAFF) or control CD45.1 donor HSPCs. Primary transplants (1°) are denoted as solid lines, secondary transplants (2°) as dotted lines. (**C**) Microscope images of modified Wright’s stained touch preps and peripheral blood smears of age-matched control (Normal) or leukemic (MLL-AF9) tissues derived from *Hoxa*^flox/flox^ mice. Scale bar 20 µm. (**D**) Digital image of spleens removed from an age-matched control mouse (upper) and leukemic (MLL-AF9) *Hoxa*^flox/flox^ mouse (lower).

**Figure 2 cancers-11-02036-f002:**
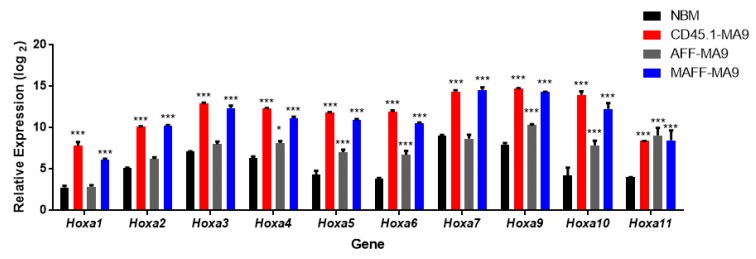
Overexpression of *Hoxa* cluster genes in MA9 leukemias. Bar chart of relative gene expression of *Hoxa* cluster genes in normal bone marrow (NBM) and MA9 leukemias derived from wild type (CD45.1-MA9), Hoxa^flox/flox^ (AFF-MA9) and MxCre^+^/Hoxa^flox/flox^ (MAFF-MA9) genetic backgrounds. The mean values from triplicate experiments are plotted. Significance as calculated by 1 way ANOVA compared to control bone marrow is denoted as * *p* ≤ 0.05; ** *p* ≤ 0.01, *** *p* ≤ 0.001.

**Figure 3 cancers-11-02036-f003:**
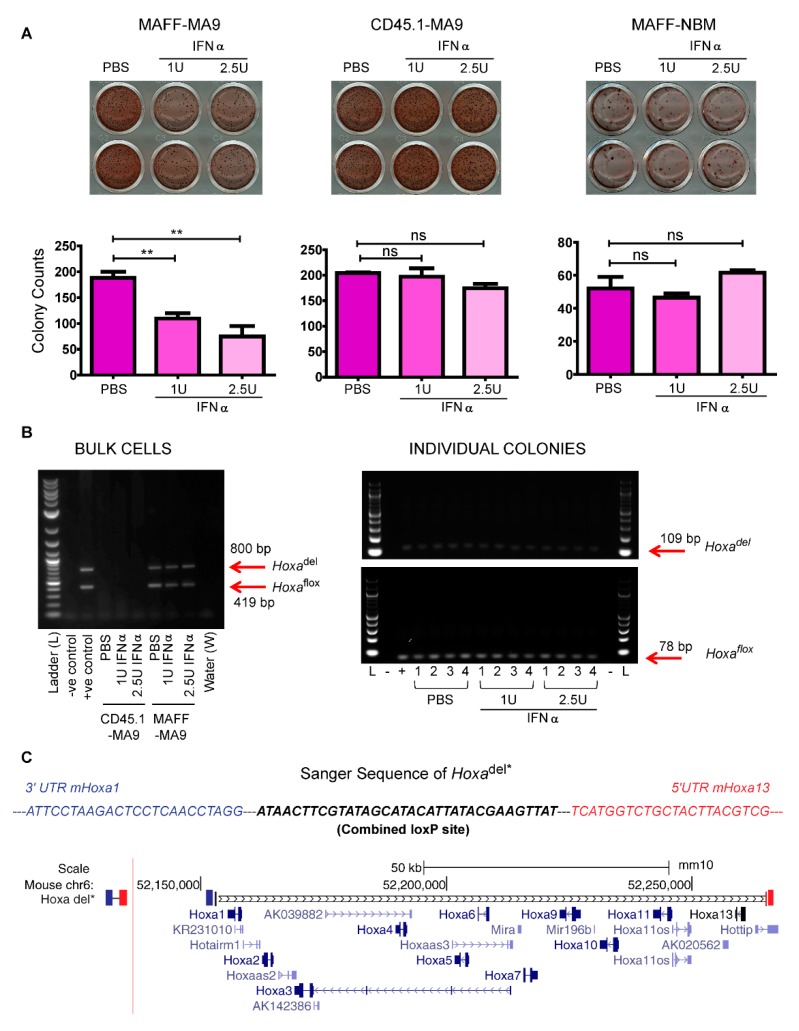
In vitro deletion of the *Hoxa* cluster in *MxCre^+^*/*Hoxa*^flox/flox^ (MAFF) derived leukemic cells. (**A**) Digital images of leukemic colonies derived from MAFF-MA9 and wild type (CD45.1-MA9) mice compared to MAFF derived normal bone marrow controls (MAFF-NBM). Colonies were treated with interferon-alpha (IFN-α) at the indicated dose units (U) or with PBS vehicle control (upper panel), counted after 7 days using the GelCount™ analyzer and plotted on a bar chart (lower panel). Not-significant (ns); ** *p* ≤ 0.01. (**B**) Digital images of electrophoresed agarose gels containing PCR amplicons of expected size (arrows) derived from bulk CD45.1 or MAFF-derived leukemic colonies (left panel) or representative MAFF-MA9-derived individual colonies (right panel). (**C**) A schematic of the aligned *Hoxa*^del^ amplicon sequence with the mouse chromosome 6 tract (UCSC). The sequencing read (−600 bp) encompassed a portion of the 3ʹ UTR of *mHoxa1* (blue), the recombined *loxP* site (black) and apportion of the 5ʹ UTR of *mHoxa13* (red).

**Figure 4 cancers-11-02036-f004:**
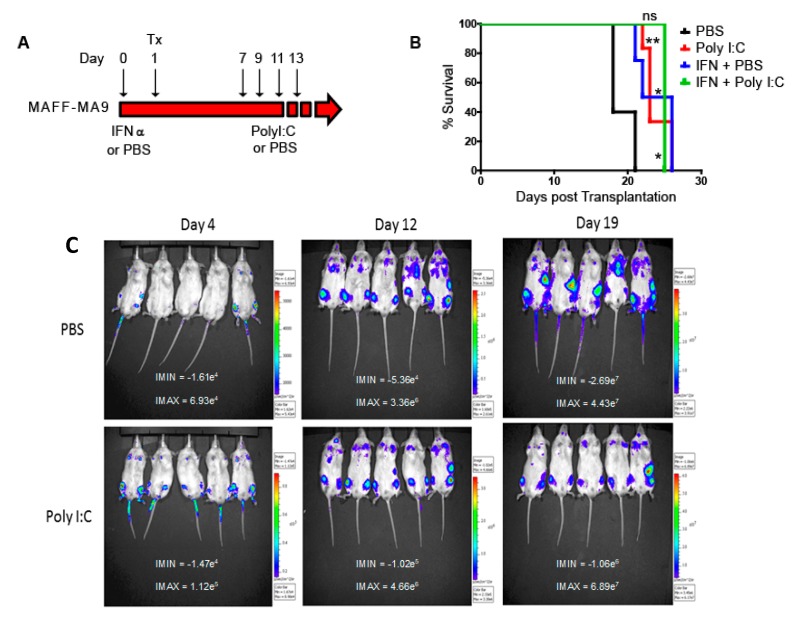
In vivo treatment of MAFF-MA9 leukemia. (**A**) Schematic of the treatment of MAFF-MA9 leukemias prior to and following transplantation into recipient mice. Interferon-alpha (IFN-α), Polyinosinic:polycytidylic acid (PolyI:C), vehicle (PBS). (**B**) Kaplan–Meier plot demonstrating survival of transplanted mice receiving MAFF-MA9 leukemia cells following treatment. * *p* ≤ 0.05; ** *p* ≤ 0.01. (**C**) In Vivo Imaging Systems (IVIS) derived images of luciferase expressing MAFF-MA9 transplanted into NOD-scid IL2rγnull (NSG) recipient mice. Mice were treated with PBS or PolyI:C and live images taken as indicated (*n* = 5 per group).

**Figure 5 cancers-11-02036-f005:**
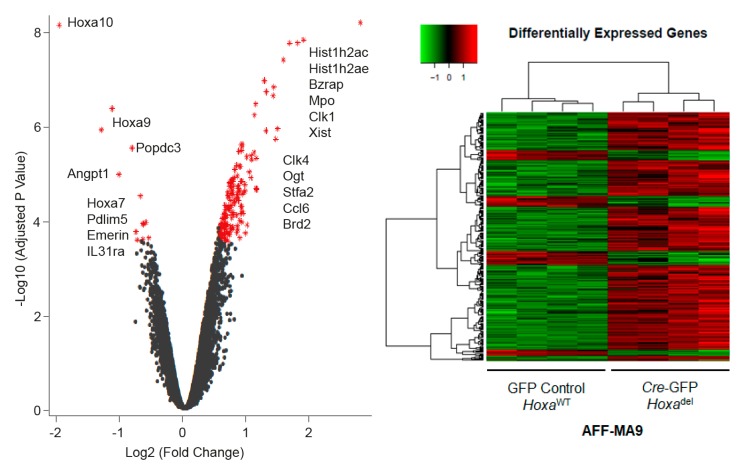
Differential gene expression in *Hoxa*^del^ MA9 cells. **Left panel**; Volcano plot of significance for each of 24,326 probes from Cre-GFP treated (*Hoxa*^del^) AFF-MA9 cells. Negative log10 (Adjusted P-value) versus log2 (fold change) compared to GFP-treated AFF-MA9 cells. Genes with –log10 (Adjusted P-value) > 2 and log2 (fold change) > 0.5 (annotated and highlighted in red) were considered differentially expressed. **Right panel**; sample transcripts induced or repressed at log2 (fold change) of 0.5 or more (*p* ≤ 0.05) were subjected to unsupervised hierarchical agglomerative clustering by treatment with Cre based on Euclidean distance and linkage.

**Figure 6 cancers-11-02036-f006:**
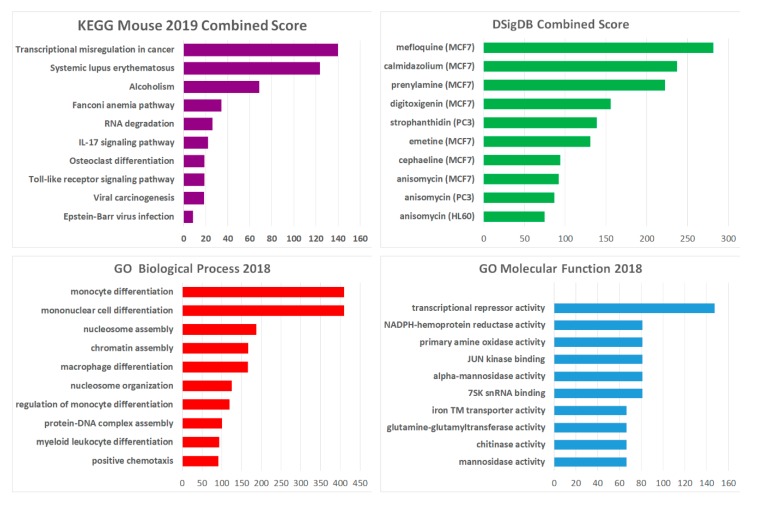
Gene set enrichment analysis of the *Hoxa*^del^ signature. Summary bar charts demonstrating association of the *Hoxa*^del^ signature with databases including Kyoto Encyclopedia of Genes and Genomes (KEGG) pathways, Gene Ontology (GO) of Molecular Function and Biological processes and the NCBI drug signatures database for gene set analysis (DSigDB) using Enrichr [30,31]. Combined scores (based on the log Fisher exact test *p*-value and *z*-score for deviation from expected rank) are plotted against the key pathway, process or drug interactions.

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
