# Peer review of "Pathways, Processes, and Candidate Drugs Associated with a Hoxa Cluster-Dependency Model of Leukemia"

_cancers, 2019, doi:10.3390/cancers11122036_

Round 1

Reviewer 1 Report

Kettyle et al present a well written and well referenced manuscript. The authors describe the effect of reduced expression of Hoxa genes on the maintenance of a mouse leukemia driven by the MLL-AF9 fusion protein.

One major factor that impacts the ability to interpret the data presented in this study is the apparent leakiness of the Mx1-Cre. The similar but differing systems that are used to delete the Hoxa locus also confounds the interpretation and more explanation is required.

Major Comments:

Can the authors propose any explanation as to why the Mx1-Cre has so much background activity in this system? Have the authors confirmed that this is indeed due to the Mx1-Cre by assessing the expression of the floxed allele in AFF-MA9 cells? Furthermore, can you demonstrate full deletion of the Hoxa cluster in any cells following either the IFNa treatment or Poly(I:C)? In MAFF-NBM cultures, for example?

To generate leukemias, stem/ progenitor BM cells were transformed with MLL-AF9, serially replated in methylcellulose and transplanted into mice. In figure 1b, the authors demonstrate a dramatic decrease in latency with one passage of the MA9 leukemia. In the text, it is indicated that primary leukemias were generated in mice following 3-5 passages through methylcellulose.  Do the authors predict that serial replating similarly affects the aggressiveness of the leukaemia? It is not clear if the leukemias shown in 1b have been cultured in methylcellulose for the same number of passages. Please clarify.

Presumably, based on data shown in figures 3b and 3c, MAFF cells have some level of deletion of the Hoxa However, there is no significant difference reported between AFF and MAFF tumours in vivo. Have any additional clones been tested for the different genetic backgrounds? Can the authors comment of the lack of difference between the AFF and MAFF cells in vivo?

Again, assuming that MAFF cells (even prior to IFNa treatment) have some background level of deletion of the Hoxa cluster, for the data presented in Figure 2: Why are the Hox genes not down-regulated in these cells? In contrast, AFF-AF9 cells should have similar expression levels to CD45.1-MA9 cells, however, this is not the case. Can the authors please comment on a) Why the Hox genes are not upregulated in the AFF-MA9 cells at similar levels to that seen in the CD45.1-MA9 cells, and b) why the Hox gene expression levels are not reduced in MAFF-MA9 cells?

In figure 3, treatment with IFNa is shown to reduce the colony-forming ability of MAFF-AF9 cells with no apparent effect on CD45.1-MA9 or MAFF-NBM. They further demonstrate that treatment with IFNa does not lead to further deletion of the Hoxa locus, in either bulk cultures or individual colonies. How then, does the IFN affect the growth of these colonies?

In figure 4, the authors demonstrate that pre-treatment with IFNa or poly(I:C) following engraftment of MAFF-MA9 cells increases the latency of these cells to cause a lethal leukemia. Were the authors able to demonstrate that the dose of poly(I:C) was able to delete the Hoxa locus in vivo? What recipient mice were used?

Why were NSG mice used for 4C? Is the luciferase-expressing vector different to the MLL-AF9 vector that was used for previous experiments? For the IVIS imaging, please load files together to create a combined image. The panels shown in this figure are all on different scales, making the data hard to interpret.

Is the Cre vector used in figure 5 more efficient at deleting the locus? It is evident that Hox genes are being down-regulated using the vector-based system. Please provide evidence of efficient deletion of the locus, as this is critical for the interpretation of the bioinformatic analysis.

The identification of anisomycin as a targeted therapy for MLL-rearranged leukemias is interesting. Is there evidence of RUNX1 playing a role in the regulation of Hox genes?

Minor comments:

Please provide higher quality images for those shown in Figure 1C. Immunophenotyping of the leukemias is mentioned in the methods but not shown. What was the immunophenotype of these AMLs? In figure S2 is it HOX WT/WT or HOX FL/FL? Please be consistent with nomenclature. Please provide a better image for the “individual colonies” blot shown in figure 3b. Gels have been cropped too close to the band of interest. Why are the PCR product sizes different in 3b compared to 3c? In the immunoblot shown in figure 1E, wild type MLL (MLLN) is this increased in the cell lines compared to the primary leukemia? It is hard to tell from the loading.  If it is increase, is there any possible explanation for this? Were any changes in MLL itself detectable in the gene expression data?

Author Response

Response to Reviewer 1.

Can the authors propose any explanation as to why the Mx1-Cre has so much background activity in this system? Have the authors confirmed that this is indeed due to the Mx1-Cre by assessing the expression of the floxed allele in AFF-MA9 cells? Furthermore, can you demonstrate full deletion of the Hoxa cluster in any cells following either the IFNa treatment or Poly(I:C)? In MAFF-NBM cultures, for example?

Primary leukaemias were established in irradiated (450-600 cGy) immunocompetent mice which would promote an inflammatory microenvironment that could activate the Mx1 interferon response element, driving Cre expression. In addition, leukemia initiation is associated with a pro-inflammatory response that includes upregulation of IL-1, TNF and IFN-gamma (Carey et al Cell Reports 18;13 P3204-3218, 2017) that may prematurely stimulate Mx1-Cre in our model. This is supported by the lack of MxCre ‘leakage’ in our previously reported normal hematopoietic (MAFF-NBM) model (Lebert-Ghali et al Blood. 2016 Jan 7;127(1):87-90. doi: 10.1182/blood-2015-02-626390)

To generate leukemias, stem/ progenitor BM cells were transformed with MLL-AF9, serially replated in methylcellulose and transplanted into mice. In figure 1b, the authors demonstrate a dramatic decrease in latency with one passage of the MA9 leukemia. In the text, it is indicated that primary leukemias were generated in mice following 3-5 passages through methylcellulose. Do the authors predict that serial replating similarly affects the aggressiveness of the leukaemia? It is not clear if the leukemias shown in 1b have been cultured in methylcellulose for the same number of passages. Please clarify.

To clarify, secondary leukemias were generated by directly transplanting primary leukemia cells. A sentence to that effect has been inserted at line 82. Serial replating of the leukemia was not done.

Presumably, based on data shown in figures 3b and 3c, MAFF cells have some level of deletion of the Hoxa. However, there is no significant difference reported between AFF and MAFF tumours in vivo. Have any additional clones been tested for the different genetic backgrounds? Can the authors comment of the lack of difference between the AFF and MAFF cells in vivo?

No additional clones were tested. MLL-AF9 leukaemias are highly aggressive and rapidly growing with a doubling time of <24 hours therefore a reduction in burden even by 50% may result in less than a single day extension in time to leukemia in vivo.

Again, assuming that MAFF cells (even prior to IFNa treatment) have some background level of deletion of the Hoxa cluster, for the data presented in Figure 2: Why are the Hox genes not down-regulated in these cells? In contrast, AFF-AF9 cells should have similar expression levels to CD45.1-MA9 cells, however, this is not the case. Can the authors please comment on a) Why the Hox genes are not upregulated in the AFF-MA9 cells at similar levels to that seen in the CD45.1-MA9 cells, and b) why the Hox gene expression levels are not reduced in MAFF-MA9 cells?

The initial cells that were targeted by MLL-AF9 in all three backgrounds were enriched for a heterogeneous population of hematopoietic stem and progenitor cells by design as the true cell-of-origin for MLL-AF9 is not known. Figure 2 confirms that comparisons between backgrounds (MAFF, AFF or CD45.1) must be treated with caution due to initiating/target cell differences.

In figure 3, treatment with IFNa is shown to reduce the colony-forming ability of MAFF-AF9 cells with no apparent effect on CD45.1-MA9 or MAFF-NBM. They further demonstrate that treatment with IFNa does not lead to further deletion of the Hoxa locus, in either bulk cultures or individual colonies. How then, does the IFN affect the growth of these colonies?

IFN reduces the growth of MAFF-MA9 colonies by stimulating Mx1 resulting in expression of Cre recombinase causing deletion of the Hoxa cluster. CD45.1-MA9 cells do not contain the Mx1-Cre inducible element and therefore retain the Hoxa cluster. MAFF-NBM cells are included to rule out non-specific IFN toxicity and also demonstrates less reliance on the Hoxa cluster for colony growth of normal bone marrow compared to MLL-AF9 cells.

In figure 4, the authors demonstrate that pre-treatment with IFNa or poly(I:C) following engraftment of MAFF-MA9 cells increases the latency of these cells to cause a lethal leukemia. Were the authors able to demonstrate that the dose of poly(I:C) was able to delete the Hoxa locus in vivo?

The authors have previously shown that the dose of Poly(I:C) is sufficient to delete both alleles of the Hoxa locus in vivo in the normal hematopoiesis model (Lebert-Ghali et al Blood. 2016 Jan 7;127(1):87-90. doi: 10.1182/blood-2015-02-626390) but were unable to observe this in vivo in the MLL-AF9 model with at least one allele being retained, indicative of MLL-AF9 dependency on the Hoxa cluster.

What recipient mice were used?

NOD-scid IL2rγnull (NSG) mice were used as recipients.

Why were NSG mice used for 4C? Is the luciferase-expressing vector different to the MLL-AF9 vector that was used for previous experiments? For the IVIS imaging, please load files together to create a combined image. The panels shown in this figure are all on different scales, making the data hard to interpret.

NSG mice were used to reduce potential pro-inflammatory responses associated with irradiated recipients. The MLL-AF9 coding region was exactly the same as for the previous experiments and cloned upstream of a luciferase cassette (pSLIEW) as stated on lie 310.

The bioluminescence tracing was used primarily as a tool to verify leukemia progression and intervene prior to animal death rather than a precise quantification tool. Mice were imaged as a group for comparison that resulted in scaling, rather than for individual photons/second measurements.

Is the Cre vector used in figure 5 more efficient at deleting the locus? It is evident that Hox genes are being down-regulated using the vector-based system. Please provide evidence of efficient deletion of the locus, as this is critical for the interpretation of the bioinformatic analysis.

The efficiency of Cre is likely to be the same in all contexts. However the ability to purify cells based on high GFP allowed for enrichment of Cre-treated cells.

Evidence of efficient deletion is now included in Figure S2 with associated text “Methylcellulose-based colony formation and PCR demonstrated efficient deletion of the Hoxa locus (Figure S4)” included at line 191.

The identification of anisomycin as a targeted therapy for MLL-rearranged leukemias is interesting. Is there evidence of RUNX1 playing a role in the regulation of Hox genes?

The t(3;21) induced RUNX-EVI1 is associated with activation of HOXA associated genes and HOXA9 (Loke et al Cell Rep. 2017 May 23;19(8):1654-1668).

Minor comments:

Please provide higher quality images for those shown in Figure 1C. Immunophenotyping of the leukemias is mentioned in the methods but not shown. What was the immunophenotype of these AMLs? In figure S2 is it HOX WT/WT or HOX FL/FL? Please be consistent with nomenclature. Please provide a better image for the “individual colonies” blot shown in figure 3b. Gels have been cropped too close to the band of interest. Why are the PCR product sizes different in 3b compared to 3c? In the immunoblot shown in figure 1E, wild type MLL (MLLN) is this increased in the cell lines compared to the primary leukemia? It is hard to tell from the loading. If it is increase, is there any possible explanation for this? Were any changes in MLL itself detectable in the gene expression data?

Images were copied as PDFs and applied to the Cancers template and will be adjusted as necessary following discussion with the editor(s).

Immunophenotyping for the primary leukemias and cell lines is now included in new Table S1 and referred to in the main text on line 81-82.

Nested PCR was used to identify individual colony bands (3C) hence the size difference. A sentence has been inserted at line 325 and reference to the supplemental new Table S3 at line 366.

Figure 1E will be removed.

No changes in MLL itself was detected in the gene expression data.

Reviewer 2 Report

In this manuscript, Kettyle LM et al., present how HOXA expression drives leukaemia progression in two different in vivo models (myeloid and acute leukaemias). They also explored how HOXA depletion alters the transcriptomic profile in leukaemic cells.

This manuscript could be divided in two different parts, the in vivo analysis and the gene expression study. Although the study looks interesting and very well conducted, both parts are not well connected.

Regarding to the first part, authors compared in Figure 2 the leukaemic models vs NBM; however, there are no differences between "control CD45-1 MA9" and AFF-MA9. Also, in most of HOXA genes (except Hoxa11) MAFF-MA9 clearly presents similar levels to NBM and lower than CD45-1 and AFF. Authors should expand this result and discuss this further.

Regarding to the colony formation assay (Figure 3), is there any difference between MAFF and CD45 cells? Seems that there is not. If so, would the authors expected more colonies in MAFF than CD45 cells?

Due to MAFF results presented in Figure 2, would the authors expect that IFN treatment influences on Hoxa 4, 5, 6, 9, 10, or 11 (which are upregulated compared to NBM)?

The transcriptomic analysis performed in the second part of the manuscript looks interesting and I do not have further issues regarding to this part.

Overall, I think the manuscript looks promising and of interest to Cancers readers. 

Author Response

Response to Reviewer 2.

Regarding to the first part, authors compared in Figure 2 the leukaemic models vs NBM; however, there are no differences between "control CD45-1 MA9" and AFF-MA9. Also, in most of HOXA genes (except Hoxa11) MAFF-MA9 clearly presents similar levels to NBM and lower than CD45-1 and AFF. Authors should expand this result and discuss this further.

Figure 2 is Hoxa gene expression in leukemias (MA9) generated in 3 different backgrounds (CD45.1, AFF and MAFF) compared to normal bone marrow (NBM) and NOT to each other. As expected, not all Hoxa genes are activated by MA9, but many are in all 3 backgrounds to statistical significance.

Regarding to the colony formation assay (Figure 3), is there any difference between MAFF and CD45 cells? Seems that there is not. If so, would the authors expected more colonies in MAFF than CD45 cells?

As both MAFF-MA9 and CD45.1-MA9 are leukemic cells it is not expected to see a difference in colony formation in the presence of vehicle (PBS) but the difference observed following IFN treatment reflects dependency on Hoxa expression.

Due to MAFF results presented in Figure 2, would the authors expect that IFN treatment influences on Hoxa 4, 5, 6, 9, 10, or 11 (which are upregulated compared to NBM)?

Figure 2 is not related to IFN treatment, but to MLL-AF9 (MA9) expression which regulates Hoxa subsets.

Reviewer 3 Report

In the following paper "Pathways, Processes and Candidate Drugs Associated with a Hoxa Cluster-dependency model of Leukemia", Kettyle et al., described a series of in vivo and functional assays that demonstrate the dependencies of Hoxa cluster for leukemogenesis in an MLL-AF9 acute myeloid leukemia model. Results are convincing and the paper is well suited for publication in cancers.

Below some comments in order to clarify data presentation:

1) This reviewer strongly suggests to quantifying the emission of photons in bioluminescence assay, generally presented as Photons/Second. This approach is especially important because the difference between the curves is not too impressive (five or six days);

2) GSEA analysis is very interesting, however, Figure 6 can be improved, please provide GSEA graphs of selected pathways and clarify what means the abscissa axis;

3) The paragraph between lines 271 - 275 in discussion seems too speculative since the bone marrow microenvironment was not evaluated in the paper. I would like to suggest to discuss the function of Hoxa genes in the context of mutation of well recognized epigenetic genes, such as ASXL1 and DNMT3A.

4) In Fig 6 Legend, please replace "correlation" (a statistical test for continuous variables) for "association"; 

5) It would be interesting if the authors could test the cytotoxic effects of some drugs that they identified as potentially related to Hoxa pathways;

6) Please clarify what kind of radiation was used (X-ray, Gamma...?), this is important to understand the effects of the dose. As well as, please refer the immunophenotyping strategy;

7) It would be of interest if the authors provided a statement for publicity of data;

8) Please verify the quality of images across the figures, such as for example, Fig. 3B may be better-visualized adjusting contrast.

Author Response

Response to Reviewer 3.

1) This reviewer strongly suggests to quantifying the emission of photons in bioluminescence assay, generally presented as Photons/Second. This approach is especially important because the difference between the curves is not too impressive (five or six days);

MLL-AF9 leukaemias are highly aggressive with a doubling time of <24 hours therefore a 50% reduction in burden may only result in a 1 day extension in time to leukemia. However, as in patients, leukaemia burden does not equate to time to leukemia. The bioluminescence tracing was used primarily as a tool to verify leukemia progression and intervene prior to animal death rather than a precise quantification tool. Mice were imaged as a group for comparison rather than for individual photons/second measurements without scaling. The results presented are significant at the levels stated.

2) GSEA analysis is very interesting, however, Figure 6 can be improved, please provide GSEA graphs of selected pathways and clarify what means the abscissa axis;

The authors agree with other Reviewers that the GSEA analysis from Enrichr is sufficient, with limitations, for the associations drawn.

3) The paragraph between lines 271 - 275 in discussion seems too speculative since the bone marrow microenvironment was not evaluated in the paper. I would like to suggest to discuss the function of Hoxa genes in the context of mutation of well recognized epigenetic genes, such as ASXL1 and DNMT3A.

The authors believe the statement that “In combination with Hoxa7, a9, and a10, reduced expression of endothelial/adhesion molecules (Heg1 and Angptl1) may be associated with altered Hoxadel cell-niche interactions” is appropriate in this context.

The t(11q;23) translocation is sufficient for leukemia initiation and progression and is not associated with additional mutations such as ASXL1 and DNMT3A as reported by The Cancer Genome Atlas (Miller et al., N Engl J Med. 2013 Oct 10;369(15):1473.

4) In Fig 6 Legend, please replace "correlation" (a statistical test for continuous variables) for "association";

Done.

5) It would be interesting if the authors could test the cytotoxic effects of some drugs that they identified as potentially related to Hoxa pathways;

Such experiments are multi-faceted as to the model(s) of choice and use as monotherapies or in combination which is outside the scope of this submission.

6) Please clarify what kind of radiation was used (X-ray, Gamma...?), this is important to understand the effects of the dose. As well as, please refer the immunophenotyping strategy;

Gamma radiation was used from a Caesium-137 source.

7) It would be of interest if the authors provided a statement for publicity of data;

All data pertaining to the manuscript (including pertinent gene expression data) has been included as Supplemental.

8) Please verify the quality of images across the figures, such as for example, Fig. 3B may be better-visualized adjusting contrast.

Images were copied as PDFs and applied to the Cancers template and will be adjusted as necessary following discussion with the editor(s).

Reviewer 4 Report

Here, Kettyle and coworkers tried to understand how MLLr leukemia is dependent on Hoxa expression. Although the manuscript answered questions that are relevant and would interest readers in the field, it still requires work to be considered for publication. Here are my comments:

Do the authors think that in trans expression of Hoxa cluster can initiate and help progress MLLr? Can they provide any data for that? Fig 1E shows a very high protein level of HSP90. Can the authors use a different immunoblot and quantify it? Fig 3A needs to be replaced with a better quality figure as the current figure is not high resolution. Fig 4C scaling needs to be changed. The authors need to validate at least a few genes that are differentially expressed in Hoxadel MA9 cells. Fig 6, the gene set enrichment analysis of the Hoxadel signature is an interesting finding. However, it is correlative unless any of the pathways are validated by biochemical assays. Without validation, the authors should at least modify the title, as it does not paint a correct picture of the current manuscript.

Author Response

Response to Reviewer 4.

Do the authors think that in trans expression of Hoxa cluster can initiate and help progress MLLr? Can they provide any data for that?

The Hoxa cluster is a direct target of MLL-AF9 rather than functioning in trans.

Fig 1E shows a very high protein level of HSP90. Can the authors use a different immunoblot and quantify it?

The immunoblot will be removed.

Fig 3A needs to be replaced with a better quality figure as the current figure is not high resolution.

Images were copied as PDFs and applied to the Cancers template and will be adjusted as necessary following discussion with the editor(s).

Fig 4C scaling needs to be changed.

The bioluminescence tracing was used primarily as a tool to verify leukemia progression and intervene prior to animal death rather than a precise quantification tool. Mice were imaged as a group for comparison that resulted in scaling, rather than for individual photons/second measurements.

Fig 6, the gene set enrichment analysis of the Hoxadel signature is an interesting finding. However, it is correlative unless any of the pathways are validated by biochemical assays. Without validation, the authors should at least modify the title, as it does not paint a correct picture of the current manuscript.

The authors believe the term Associated in the Title rather than Correlated is appropriate:

Reviewer 5 Report

The manuscript by Kettyle and co-workers demonstrates that Hoxa cluster is critically in the maintenance and establishment of aggressive MA9 leukemia. Authors have shown that Cre-recombinase deletion of Hoxa locus resulted in loss of leukemic phenotype in vitro and increased survival in vivo. Other experiments have further confirmed the Hoxa locus leukemic dependency and bioinformatics analysis identified a number of FDA-approved drugs with potential application in clinical sets. This is a very interesting manuscript that brings light to a new potentially druggable target in leukemia. There are still some controls that must be presented on the establishment and validation of conditional Hoxa leukemia models. Authors should confirm the capacity of conditional HoxA−/−progenitors to repopulate. The results presented on figure 1C, 1D and 1E are not enough to clarify this aspect. Furthermore, the same control is needed in data presented in figure 4.

Author Response

Response to Reviewer 5.

Authors should confirm the capacity of conditional HoxA−/−progenitors to repopulate.

The authors have previously shown repopulation of conditional Hoxa-/- progenitors using the exact same Hoxaflox/flox/MxCre mouse background in the normal hematopoiesis setting (Lebert-Ghali et al Blood. 2016 Jan 7;127(1):87-90. doi: 10.1182/blood-2015-02-626390) and referenced this in the submission.

The results presented on figure 1C, 1D and 1E are not enough to clarify this aspect. Furthermore, the same control is needed in data presented in figure 4.

Results presented in Figure 1C, 1D, 1E are from Hoxaflox/flox background as an exemplar, but all backgrounds give rise to the same degree of leukaemia in terms of splenomegaly, and tissue infiltration. Figure 1E was only a confirmatory blot of the 220 kD MLL-AF9 protein and will be removed. Figure 4 is of traceable MLL-AF9 leukemia in the Hoxaflox/flox/MxCre background with the appropriate vehicle (PBS) controls.

Round 2

Reviewer 1 Report

Dear Kettyle et al,

Thank you for your responses and clarification.  Please see below a couple of follow-up questions/ comments:

Were the cell populations analysed in Figures 2 and 3 derived from transduced bone marrow cells or bulk leukemia? That is, where they CFU-GM that had been serially passaged, or whole bone marrow from a leukemic mouse (derived from CFU clones)? As the authors have stated in their response: “serial replating of the leukemia was not done” however, the cells were serially passaged prior to transplantation (between 3-5 passages). Would this not have an effect on the aggressiveness of the cells? The difference between 3 and 5 re-platings could be significant. Perhaps a slightly more detailed explanation of the methodology would help here. As the authors have suggested, background deletion of the Hoxa cluster could have occurred in vitro following inflammation due to the transplantation. Again, a quick sentence explaining that the colonies shown in Figure 3 are not from the initial serial- replating but are the “leukemic” cells from a transplanted population, would be very useful here. For figure 3, you have stated that further activation of the Cre recombinase was not achieved by treatment with IFNa and no further deletion of Hoxa was observed. How then, is there any effect on cell growth? I appreciate that the IFN activates the Mx1-Cre but as the deletion has already occurred in vivo and no further deletion is possible (presumably due to an absolute requirement of Hox gene expression) and you have ruled out IFN or Cre toxicity, what is your interpretation?  Is there in fact further deletion (this may be apparent in your bulk analysis) and are you able to more accurately quantify this? Given that multiple reviewers have requested a revised figure 4c, this reviewer is surprised at the reluctance of the authors to provide a grouped/ scaled image. This is very straightforward and should always be done when presenting IVIS imaging data. Showing multiple images on different scales over time does not show progression. In fact, given the aggressiveness of this leukemia, loading the images as a group is likely to improve the figure and give a more striking difference between D12 and D19. In addition, it appears from the KM plot that not all of the PBS group were alive on D19. Please confirm the timing of this imaging.

Author Response

The authors found it difficult to address the Reviewers "couple" of follow up questions and comments (which are actually 6) in point by point as they do not refer to specific regions of the document or tracked by line number etc.

To clarify:

1.  The cell populations analysed in Figures 2 and 3 were derived from bulk leukemia.

2.  The serial passaging of transduced BM cells was done to select for higher proliferating cells as is routinely done to generate such models.  Line 75 clearly states:

"Extended serial re-plating in methylcellulose selected for transformed HSPCs with high proliferation potential".

3.  The authors have suggested, background deletion of the Hoxa cluster could have occurred in vitro following inflammation due to the transplantation.

The authors are looking for a reference as to where this was stated, but if this refers to the Figure 3 legend, an inflammatory response could have occurred at multiple points in the generation of MAFF-MA9 primary cells either (a) the time of transduction of normal HSPCs with virus (in vitro) or (b) the time of engraftment of the leukemic cells into the bone marrow (in vivo) or indeed (c) at the time the cells were extracted from the bone marrow and placed in culture.  Any of the above would support the observation made.

4.  The authors state on Line 139/140 that "background Cre activation that was not markedly altered by IFNα treatment in bulk cells".  gDNA PCR is non-quantitative and even small changes in HOXA gene expression not detected by gDNA PCR (e.g. individual siRNA) is associated with reduced colony formation in MLL-AF9 leukemias. 

5.  The authors believe  that the gDNA PCR method is appropriate to demonstrate the deletion of the cluster rather than using e.g. QPCR to examine the expression of individual genes.

6. The transplanted leukemias in 4b and 4c are NOT the same. Figure 4b and 4c are clearly labelled: (B) Kaplan-Meier plot demonstrating survival of transplanted mice receiving MAFF-MA9 leukemia cells following treatment. *p ≤ 0.05; **p ≤ 0.01. (C) In Vivo Imaging Systems (IVIS) derived images of luciferase expressing MAFF-MA9 transplanted into NOD-scid IL2rγnull (NSG) recipient mice.

Reviewer 4 Report

1. In the previous review, I asked "Do the authors think that in trans expression of Hoxa cluster can initiate and help progress MLLr? Can they provide any data for that?"

The authors responded: The Hoxa cluster is a direct target of MLL-AF9 rather than functioning in trans.

The question was whether Hoxa cluster or MLL-AF9 has the potential to drive MLLr. I do not think the reply answers the question. If they do not wish to add any data at this point, at least making it a discussion point would be useful

2. The previous review had asked the authors to change Fig1E. "Fig 1E shows a very high protein level of HSP90. Can the authors use a different immunoblot and quantify it?"

The authors responded: "The immunoblot will be removed." However, blot still remains.

3. It is impossible to understand the scaling from Fig4C.

4. The title still does not properly showcase the manuscript.

Author Response

1.  The question was whether Hoxa cluster or MLL-AF9 has the potential to drive MLLr.

MLL-AF9 is MLLr and in this model MLL-AF9 (MLLr) is driven by a strong retroviral promoter.  MLLr is an extremely rare rearrangement brought about by genomic instability.  There is no evidence of the Hoxa cluster or MLL-AF9 driving MLLr in any reports of these models over the last decade and more.

2. Apologies, the blot will be removed.

3. The authors acknowledge this, due to the journal template. A better quality or annotated version of Figure 4c will be uploaded in the final version following discussions with the Editor/Journal staff.

4. The authors respectively agree with the other 4 Reviewers.